# Polyamines and Their Biosynthesis/Catabolism Genes Are Differentially Modulated in Response to Heat Versus Cold Stress in Tomato Leaves (*Solanum lycopersicum* L.)

**DOI:** 10.3390/cells9081749

**Published:** 2020-07-22

**Authors:** Rakesh K. Upadhyay, Tahira Fatima, Avtar K. Handa, Autar K. Mattoo

**Affiliations:** 1Sustainable Agricultural Systems Laboratory, United States Department of Agriculture, Agricultural Research Service, Henry A. Wallace Beltsville Agricultural Research Center, Beltsville, MD 20705-2350, USA; rakesh.upadhyay@usda.gov; 2Center of Plant Biology, Department of Horticulture and Landscape Architecture, Purdue University, W. Lafayette, IN 47907, USA; fatimat@purdue.edu (T.F.); ahanda@purdue.edu (A.K.H.)

**Keywords:** cold-stress, heat-stress, polyamines-biosynthesis, polyamines-catabolism, putrescine (PUT), spermidine (SPD), spermine (SPM), tomato

## Abstract

Polyamines (PAs) regulate growth in plants and modulate the whole plant life cycle. They have been associated with different abiotic and biotic stresses, but little is known about the molecular regulation involved. We quantified gene expression of PA anabolic and catabolic pathway enzymes in tomato (*Solanum lycopersicum* cv. Ailsa Craig) leaves under heat versus cold stress. These include *arginase*
*1* and *2*, *arginine decarboxylase 1* and *2*, *agmatine iminohydrolase*/*deiminase 1*, *N-carbamoyl putrescine amidase*, two *ornithine decarboxylases*, three *S-adenosylmethionine decarboxylases*, two *spermidine synthases*; *spermine synthase*; *flavin-dependent polyamine oxidases* (*SlPAO4-like* and *SlPAO2*) and *copper dependent amine oxidases* (*SlCuAO* and *SlCuAO-like*). The spatiotemporal transcript abundances using qRT-PCR revealed presence of their transcripts in all tissues examined, with higher transcript levels observed for *SAMDC1*, *SAMDC2* and *ADC2* in most tissues. Cellular levels of free and conjugated forms of putrescine and spermidine were found to decline during heat stress while they increased in response to cold stress, revealing their differential responses. Transcript levels of *ARG2*, *SPDS2*, and *PAO4-like* increased in response to both heat and cold stresses. However, transcript levels of *ARG1/2*, *AIH1*, *CPA*, *SPDS1* and *CuAO4* increased in response to heat while those of *ARG2, ADC1,2, ODC1, SAMDC1,2,3, PAO2* and *CuPAO4-like* increased in response to cold stress, respectively. Transcripts of *ADC1,2*, *ODC1,2*, and *SPMS* declined in response to heat stress while *ODC2* transcripts declined under cold stress. These results show differential expression of PA metabolism genes under heat and cold stresses with more impairment clearly seen under heat stress. We interpret these results to indicate a more pronounced role of PAs in cold stress acclimation compared to that under heat stress in tomato leaves.

## 1. Introduction

Crop losses due to different abiotic stressors are impediments to sustainable food production. Extreme temperatures (cold, freezing or heat), drought, and salinity are some of the known serious stressors [1,2,3,4,5,6,7,8]. Each of these stressors impose specific complexity in biological and genetic response of a plant and therefore to finding solutions to mitigate them. It is therefore important to develop an understanding of the complex response(s) inherent to each abiotic stress and catalog/delineate these for each crop plant to enable translational research. A number of reviews have focused on several aspects of plant abiotic stresses highlighting the serious need for delineating the processes involved and possible ways to mitigate them by utilizing ‘antistress effectors’, developing new genetic approaches, and creating new resilient crops [4,6,7,8,9]. Thus, possible players responsive to different abiotic stressors include, for instance, transcription factors (*b-ZIP, ERF/AP2 family, DOF, HD-ZIP, MYB, NAC, WRKY*, and *Zn-finger*) and other genes (*CDPKs, HAP/CAAT, HSPs-LEA family, MAPKKK*) [9,10,11,12,13,14], plant hormones (abscisic acid—ABA, salicylic acid—SA, methyl-jasmonate—MeJA, ethylene—ETH, polyamines—PAs [13,15,16], genes related to secondary metabolism [17], specific proteins (including dehydrins, heat shock proteins—HSPs, late embryogenesis abundant proteins—LEA), and osmolytes (proline/trehalose/sugars, glycine betaine) [18]. Similarly, cold temperatures can negatively impact growth and metabolism of some species such as broccoli [5], or tomato [19]. Heat/cold acclimation and tolerance impact an array of biochemical, molecular and metabolic processes in a sequential manner [20,21,22,23].

The ubiquitous PAs attracted the attention of biologists in the early 1940s as cellular amines that have a role in the biosynthesis of secondary metabolites, particularly related to the diamine putrescine [24], and later as possible antistress molecules [25,26,27]. PAs are now known to be part and parcel of plants’ life cycles ranging from cell division and differentiation, root growth and development, flower development, senescence and programmed cell death (PCD), DNA synthesis, gene transcription, cell wall loosening to fruit maturation and ripening, and plant stress tolerance [11,28,29,30,31,32,33,34,35,36]. The abundant PAs in plants include putrescine (Put), spermidine (Spd) and spermine (Spm), while the less abundant ones include cadaverine, thermospermine, norspermidine and norspermine [27]. PAs as osmo-protectants are considered to protect plants against adverse environmental conditions [11,28,31,36,37], notably drought and salinity in various crops and crop models [11,31,38,39,40,41]. These roles for PAs gained further strength from experiments carried out using a variety of agronomic and model plants including tobacco, rice, tomato, Arabidopsis, pear and potato which were engineered to accumulate PAs by overexpressing either S-adenosylmethionine (SAM) decarboxylase (showing tolerance against salt, osmotic and heat stresses) [31,42], or Spd synthase (showing tolerance against drought, salt and oxidative stresses) [31,43,44,45,46,47].

Plants utilize both ornithine and arginine biosynthesis pathways for PAs except for Arabidopsis, in which an ornithine decarboxylase (*ODC)* encoding gene(s) is yet to be found [36]. Arg is converted to Put by Arginine decarboxylase (ADC; EC 4.1.1.19) while Ornithine decarboxylase (ODC; EC 4.1.1.7) converts Orn to Put [36,48]. In addition, the activity of arginase (ARG; EC 3.5.3.1), which catalyzes Arg to ornithine, balances both pathways. Biosynthesis of higher PAs, Spd and Spm, starting from Put is catalyzed by spermidine synthase (SPDS; EC 2.5.1.16) and spermine synthase (SPMS; EC 2.5.1.22), respectively. The decarboxylation of SAM by S-adenosylmethionine decarboxylase (SAMDC; EC 4.1.1.50) adds amino propyl groups in a sequential manner to Put and Spd for the synthesis of Spd and Spm, respectively. PAs are also sequentially back-converted, from Spm to Spd to Put under particular situations, processes that involve PA oxidase (PAO; EC 1.5.3.11) and copper-containing amine oxidase (CuAO; EC 1.4.3.6). Such interconversion of PAs is thought to result in tight regulation of PAs homeostasis [49,50,51]. Both PAO and CuAO are mainly localized in cell walls and their role in PA interconversion also results in the production of the signaling molecule H_2_O_2_ as a byproduct [39,52,53,54,55]. Genome-wide identification of PA biosynthetic pathway genes [56] as well as the identification of polyamine oxidases [57] in tomato has been accomplished.

The processes by which the polyamine biosynthesis/catabolism pathways are modulated in response to different abiotic stresses are yet to be fully determined. To address this, we identified 18 tomato gene sequences involved in PA synthesis and catabolism and utilized the genome-wide transcriptome-qPCR-specific PA metabolite approach to identify PA metabolic pathway genes in the vegetative tissue of tomato. We also determined expression patterns of PA biosynthesis and catabolism genes in response to two major temperature-based abiotic stresses, viz., heat versus cold, and quantified free, conjugated and bound forms of PAs. Here we demonstrate an opposite response of tomato vegetative tissue to heat versus cold stress, highlight the patterns of PAs in each of these two abiotic stresses and their coordination with the transcription of specific PA biosynthesis and catabolism genes.

## 2. Material and Methods

### 2.1. Plant Growth Conditions

Wild type tomato (*Solanum lycopersicum* cv. Ailsa Craig) plants were grown in a temperature-controlled greenhouse under natural light conditions. After transplanting, four-week-old plants were harvested for vegetative tissues—leaves, stem, flowers and roots. Seedling samples were collected from 8–10-day-old germinated seeds with two cotyledonary leaves. All the harvested samples were immediately frozen in liquid nitrogen and stored at −80 °C until used.

### 2.2. Abiotic Stress Treatments

Heat stress treatment was carried out as described earlier [58]. Briefly, heat treatment included exposure of four-week-old (28 DAT) (days after transplanting) tomato plants to 42 °C for 24 h in a growth chamber (EGC Corp., Chagrin Falls, OH, USA) with a day/night (16/8 h) photoperiod, at 42 °C ± 1 °C and relative humidity (RH) of 50–70%. Leaf samples were collected from the heat-treated plants at 0, 0.5, 1, 6, 12 and 24 h. Harvested samples were immediately frozen in liquid nitrogen and stored at −80 °C until used.

Cold stress treatment was carried out as described earlier [58]. Briefly, cold treatment was given to four-week-old (28 DAT) tomato plants at 4 °C for 24 h in a walk-in cold room with 200 μmol/m^2^ light and 16/8 h photoperiod. Leaf samples were collected thereafter at 0, 1, 6, 12 and 24 h. The remaining process was the same as above for heat stress [58].

### 2.3. Identification and Extraction of Gene Sequences Encoding for PA Metabolism

A total of 18 genes [arginase (SlARG1, 2); arginine decarboxylase (SlADC1, 2); agmatine iminohydrolase/deiminase (SlAIH); N-carbamoyl putrescine amidase (SlCPA); ornithine decarboxylase (SlODC1, 2); S-adenosylmethionine decarboxylase (SlSAMDc1, 2, 3); spermidine synthase (SlSPDS1, 2); and spermine synthase (SPMS)] encoding for polyamine biosynthesis pathway together with four catabolic genes [Flavin-dependent polyamine oxidases (SlPAO4-like and SlPAO2) and copper-dependent amine oxidases (SlCuAO4 and SlCuAO-like)] from various published studies and National Center for Biotechnology Information (http://www.ncbi.nlm.nih.gov/) resources were used to find the tomato homologs (Appendix A). Briefly, two approaches were employed to identify potential polyamine pathway gene homologs in tomato [59]. The first approach involved protein homology search with known PA-encoding protein sequences from Arabidopsis (*A. thaliana*) and tomato (*S. lycopersicum*). The known proteins were used as query sequences for the BLASTp program in the tomato genome [International Tomato Genome Sequencing Consortium (SGN; solgenomics.net)] and phytozome databases (http://www.phytozome.net/). The second approach included validation of retrieved protein sequences using Hidden Markov Model (HMM) analysis for critical domain presence with help of domain annotation using protein motifs library (http://pfam.janelia.org).

### 2.4. Specific Marker Genes Selected for Heat/Cold Stress

To develop an understanding of specific nature of stress responses, we also tested known stress-induced genes. Thus, small heat shock protein genes (sHSPs) *HSP17.6* (Solyc06g076560.1.1), *HSP20.0* (Solyc06g076570.1.1) and *HSP20.1* (Solyc06g076540.1.1) served as markers for heat stress while C repeat/DRE/LTRE-Binding Factors (CBFs), *CBF1* (Solyc03g026280.3.1), *CBF2* (Solyc03g124110.1) and *CBF3* (Solyc03g026270.1) were tested as cold stress marker genes. HSP sequence and primer information were the same as previously published [60], whereas *CBF1-3* sequences were extracted from tomato genome database [International Tomato Genome Sequencing Consortium (SGN; solgenomics.net) as Arabidopsis homologs].

### 2.5. Marker Gene Selection for Photosynthesis Processes and Carbon Fixation

The impact of stress responses on plant’s carbon assimilation activity was tested using known genes that encode proteins involved in photosynthesis such as nuclear-encoded light-harvesting chlorophyll a/b-binding protein genes [*LHCB1* (Solyc03g005770.1), *LHCB2* (Solyc12g006140.1), *LHCB3* (Solyc07g063600.2), *LHCB4* (Solyc09g014520.2), *LHCB5* (Solyc06g063370.2), *LHCB6* (Solyc01g105050.2)] and plastid encoded photosystem II (PSII) protein genes *psbA* and *accD*. The gene information was extracted from the tomato chloroplast genome (NC_007898.3). *LHCB1-6* gene information was extracted from the tomato genome [International Tomato Genome Sequencing Consortium (SGN; solgenomics.net) database as homologs of Arabidopsis counterparts]. The plastid *accD* gene encodes the β-carboxyl transferase subunit of acetyl-CoA carboxylase while the plastid *psbA* gene encodes the D1 protein (NC_007898.3). Expression of phosphoenolpyruvate carboxylase (*PEPC*) (AJ243417.1/Solyc07g062530.2) and isocitrate dehydrogenase (*ICDH*) (XM_010314428.2/Solyc11g011930.1.1) genes were analyzed as markers for the carbon flow during heat/cold stress.

### 2.6. RNA Extraction and Quantitative PCR Analysis

Total RNA was extracted from 100 mg finely ground tomato powder using Plant RNeasy kit (QIAGEN). RNase-Free DNase (QIAGEN) was used to remove genomic DNA followed by a cleanup with a RNeasy Mini Kit (QIAGEN). RNA samples with an *A_260/280_* ratio of 1.8–2 were subjected to agarose gel electrophoresis to ensure the presence of intact rRNA band ratios before their selection for cDNA preparation [19]. RNA (2 µg) was used for cDNA synthesis using iScript Advanced cDNA synthesis kit (Bio-Rad) followed by a 10-fold dilution for further use. Quantitative real-time PCR (qRT-PCR) was performed using Sso Advanced Universal SYBR Green Supermix (Bio-Rad) in a Bio-Rad cycler (CFX96 Bio-Rad Real Time PCR machine). PCR conditions were 95 °C for 5 min, 95 °C for 15 s, and 60 °C for 60 s (40 cycles), followed by melt curve analysis [60]. Gene expression was quantified using the ∆∆C_T_ method [61]. Two reference genes (*SlTIP41* and *SlUBI3)* were used for normalizing the expression of the target genes [62,63]. MIQE (Minimum Information for publication of Quantitative real-time PCR Experiments) guidelines were followed for quantification of genes [64]. Relative fold changes were calculated as previously described [60,65]. Primer sequences used for qRT-PCR are listed in Appendix A. Relative qRT-PCR data represent average ± standard deviation from a minimum of three independent biological replicates.

### 2.7. Quantification of Put, Spd, Spm Polyamines by High Pressure Liquid Chromatography

Freeze-dried tomato leaf material was extracted and dansylated as described previously [66] with some modifications. Briefly, 50 mg of finely ground sample was homogenized in 800 µL of 5% ice-cold perchloric acid (PCA) using a hand-held homogenizer. The remaining procedure was exactly as described previously [67]. For PAs recovery and calibration curves, authentic PA standards (Sigma-Aldrich, USA) were used as control. PAs were integrated and quantified using Millennium 4.0 from Waters Corporation. PCA-soluble, PCA-soluble hydrolyzed with HCl, and PCA-insoluble after hydrolysis samples were quantified and designated as free, conjugated, and bound forms of each PA, respectively [67].

### 2.8. Statistical, Principal Component and Cystoscope Analysis

Data were examined with Graph-Pad Prism (version 8.2.1) statistical software package and two-way ANOVA (mixed model) with recommended Geisser-Greenhouse Correction was performed for deriving statistical significance. Multiple comparison within data points was performed using Tukey test. Differences among treatments were considered significant at *p* < 0.05. PCA, correlation and Cytoscape analyses were performed using XLSTAT, Excel and Cytoscape [68] programs, respectively.

## 3. Results

### 3.1. Transcriptome and qRT-PCR Analysis Highlight Differential Expression of Gene Homologues of PA Metabolic Enzymes

PA biosynthesis and catabolic genes catalogued in tomato tissue are listed in Appendix A. These genes showed dynamic expression profile in the following order *SAMDc1*>*SlSAMDc*2>*SlADC2*>*SlSPDS1* with *SAMDc1* as the most expressed one (Figure 1A). Within the Put biosynthesis gene family, *SlARG2* was the highest expressed gene relative to *SlARG1* and its expression profile in different tissues followed the order flower>leaf>seedling>stem>root. Between the two homologues of *arginine decarboxylase* (*SlADC1* and *SlADC2)*, *SlADC2* transcript levels were higher in the tomato flower than *SlADC1*. Among the two *ornithine decarboxylases*, *SlODC2* was highly expressed in the leaf and seedling tissues while significant expression of *SlODC1* was in the flower tissue. Genes encoding for the pathway from agmatine to Put, *SlAIH* and *SlCPA*, were, comparatively, less expressed and followed the order root>stem>leaf>flower>seedling (Figure 1B).

Among the Spd and Spm biosynthesis genes, *SlSAMDc2* was highly expressed in the flower, *SlSAMDc3* prominently in the leaf>seedling>flower, while *SlSAMDc1* was significantly expressed in the roots (Figure 1C). *SlSPDS1* was prominently expressed in stem and flower, while higher expression of *SlSPDS2* was restricted to the leaf, and *SlSPMS* was the least expressed as compared to the rest of the genes (Figure 1C). Among genes encoding PA catabolic enzymes, *polyamine oxidases* and *copper amine oxidases*, *SlPAO2* was expressed at higher level in flower>root and *SlPAO4-like* in root while *SlCuAO4* was highly expressed in stem>root and *CuAO4-like* expression followed the order flower>root>stem (Figure 1D).

### 3.2. Heat Stress (42 °C)

#### 3.2.1. Time-Dependent Patterns of PA Levels (Free, Conjugated and Bound Forms) in Relation to Expression Patterns of PA Metabolic Genes in Heat-Stressed (42 °C) Tomato Leaves

Heat stress led to a decrease in the levels of free Put, notably within 0.5 h of exposure and continued to decline throughout the stress period of 24 h (Figure 2A). A similar downward trend was seen in the Spd levels which was less drastic than in Put. In contrast, Spm levels remained at the same level as the control until 1 h, then increased at 6 h and remained at that level until 24 h of treatment (Figure 2A). Like the free Put, conjugated Put and conjugated Spd levels also declined with increasing period of heat while conjugated Spm levels were mostly refractile to heat (Figure 2B). In contrast, the patterns for bound PAs in response to heat were different, with bound-Put levels being very low until 6 h of exposure and thereafter a significant increase was seen only at 24 h (Figure 2C); bound-Spd levels remained significantly more or less the same until 1 h and decreased drastically thereafter at 6–24 h while the nonresponsive pattern for the bound-Spm levels was more or less similar to conjugated-Spm (Figure 2C). Significant differences between polyamine estimations are listed in Appendix A.

In response to heat, expression of PA biosynthesis pathway genes *SlARG1/2*, *SlAIH,* and *SlCPA* started increasing at 1 h, with a major increase at 6 h which continued until 24 h in the case of *SlARG1* and *SlAIH*, but levels of *SlARG2* and *SlCPA* dropped downwards (Figure 3A,D). Expression of other PA biosynthesis pathway genes *SlADC1/2* and *SlODC1/2* were significantly downregulated in response to heat stress (Figure 3B,C). Among the three *SlSAMDC* genes, all were downregulated within 0.5 h heat exposure and their expression remained low except for *SlSAMDc3* whose expression increased significantly at 6 h of heat treatment (Figure 3E). Notably, *SlSPDS1* and *SlSPDS2* expression increased within 0.5 h exposure to heat and thereafter increased maximally by 6 h (Figure 3F). Expression of *SlSPMS* gene remained unaffected until 1 h of heat exposure, decreasing considerably at 6 h and onwards (Figure 3G). PA back pathway involves catabolism of Spd/Spm by flavin adenine dinucleotide (FAD)-dependent PA oxidases (PAO2 and PAO4-like) and of Put by copper amine oxidases (CuAO4 and CuAO4-like) [39,51,54]. Expression of *SlPAO2* remained low and unaltered by heat exposure; however, *SlPAO4-like* expression significantly increased within 0.5 h of heat exposure and remained at that level at 1 and 12 h exposure time but the increase almost doubled at 6 and 24 h of heat treatment (Figure 3H). The expression of *SlCuAO4-like* gene was minimal and did not change during heat stress period (Figure 3I) while *SlCuAO4* expression increased at 1 h, peaked at 6 h and remained lower but significant until 24 h (Figure 3I). The significance of differences between various forms of PAs are listed in Appendix A.

Heat shock protein (HSP) genes are known to be induced during heat stress. Therefore, we used *HSP17.6, HSP20.0* and *HSP20.1* as heat stress marker genes to determine changes in their transcript levels. All three HSP marker genes exhibited very strong induction within 30 min and 1 h of heat stress and declined thereafter (Figure 3J–L).

#### 3.2.2. Pearson Correlations of Free (F), Conjugated (C) and Bound (B) PUT, SPD and SPM with Each Other and with Transcript Levels of PA Metabolism, sHSPs and Other Genes during Heat Stress

Out of 72 possible combinations among free (F), conjugated(C) and bound (B) PUT, SPD and SPM, during the heat stress, 28 and 20 exhibited correlation coefficients (*r*) ≥ 0.8 or ≤ −0.8, respectively,. PUT-F showed positive (+ve) *r* with PUT-C, SPD-C and SPD-B and negative (−ve) *r* with SPM-F, SPM-C and PUT-B. SPM-F showed only +ve *r* with PUT-C and SPD-C. Interestingly, SPM-F exhibited *r* opposite to that observed for PUT-F (Appendix A). These data suggest that both PUT and SPM contribute to maintaining PA homeostasis under heat stress.

Transcript levels of PAs biosynthesis and catabolic enzyme genes exhibited differential correlation coefficients with the various forms of PAs (Appendix A). Under heat stress, PUT-F did not show +ve *r* with any of the PA metabolic genes but exhibited strong −ve *r* with *ARG1* and *ODC2*. However, SPD-F showed +ve *r* with *ODC1 and SPDS2*. SPM-F was positively associated with *ARG1/2, ODC2, AIH1, CPA* and *SPDS1* and exhibited −ve *r* with *SPMS* under the heat stress (Appendix A).

To evaluate possible interactions of different PA forms in regulating gene expression, we determined their correlation coefficient with several genes including three small HSPs (Appendix A). *HSP20.0*and *HSP20.1* exhibited strong +ve *r* with PUT-F, while the three HSP genes examined showed strong −ve *r* with SPM-F. None of the HSP genes showed either +ve or −ve association with SPD-F during the heat stress. Among the other marker genes tested, +ve *r* was obtained between *Lhcb1/2* with PUT-F, and *CDH*, *Lhcb1/5* with SPD-F. *Lhcb1/2* exhibited −ve *r* with SPM-F. Interestingly, relationship of SPM-F was generally opposite (negative) to that of Put-F and SPD-F suggesting a Yin-Yang mechanism of gene regulation among them under the heat stress (Appendix A).

#### 3.2.3. Coordinated Regulation of PA Metabolic and Heat Shock Protein Genes under Heat Stress

Except for *SPMS,* which showed −ve *r* with *ARG1,2*, *AIH1* and *CPA*, most of the PA metabolism genes exhibited +ve r among themselves during the heat stress (Appendix A). *ARG1* expression pattern coordinated with *ODC2, AIH1, CPA, SPDS1, PAO4-like* and *CUPAO4*; *ARG2* with *CPA; ADC2* with *ODC2, AMDC1, SPDS2* and *PAO2; ODC1* with *ADC2, SPDS2* and *PAO2*; OD*C2* with *CuAO4-like*; *AIH1* with *CPA, SAMDC3, SPDPAO4-like* and *CuAO4*; *CPA* with *SAMDC3, SPDS1* and *CuAO4*; *SAMDC1* with *CUAO4*; *PAO2*, *PAO4*-like; *SAMDC2* with *CuAO4-like*; *SAMDC3* with *SPMS* and *CuAO4; SPDS1* with *CuAO4; SPDS2* with *PAO2*; *PAO4-like* with *CuAo4like*; with *CuAo4like* (Appendix A).

Among the marker genes that exhibited +ve r with PA metabolic genes under heat stress included *OAT1*, *PEPC*, *ICDH* and *accD* (Appendix A). The transcript levels of three *sHSP* genes analyzed exhibited −ve r with a number of genes of PA metabolism, namely, *ARG1/2*, *AIH1*, *CPA*, *SPDS1*, and *CuAO4* (Appendix A). *ARG1,2* both exhibited −ve r with *Lhcb1-3* (Appendix A). *The Lhcb1-6* and three *sHSP* genes exhibited +ve r with each other suggesting they are coordinately regulated during heat stress. Further, *OAT1* had +ve r with *PEPC* and *accD* (Appendix A).

### 3.3. Cold Stress (4 °C)

#### 3.3.1. Time-Dependent Patterns of PA Levels (Free, Conjugated and Bound Forms) in Relation to Expression Patterns of PA Metabolic Genes in Cold-Stressed (4 °C) Tomato Leaves

In contrast to heat stress, exposure of tomato leaves to 4 °C (cold stress) initiated accumulation of free Put at 6 h, increasing further with time and reached high levels at 24 h (Figure 4A). Free Spd levels remained unchanged until 12 h, increasing slightly at 24 h (Figure 4A). Although free Spm levels maintained a similar trend as that of Spd levels, their levels were significantly much lower than Spd (Figure 4A). The levels and patterns of conjugated PAs were very similar to those seen for free PAs (Figure 4B) except that conjugated Spm levels were significantly lower than conjugated SPD levels. The concentration of bound Put was very low except it increased after 24 h (Figure 4C). The bound Spd levels remained similar until 6 h of cold treatment and exhibited an increase thereafter. The levels of bound Spm decreased significantly between 1 and 12 h, returning at 24 h to the levels similar to zero h sample (Figure 4C). Significance of differences between various forms of PAs are listed in Appendix A.

Cold stress increased the expression of PA biosynthesis gene *SlARG2* within 1 h, peaking at 12 h before declining at 24 h, whereas *SlARG1* gene levels remained similar throughout the cold treatment (Figure 5A). A trend similar to *SlARG2* was observed in the expression of other PA biosynthesis genes *SlADC1/2*, and *SlODC1* (Figure 5B,C). In contrast, the expression of *SlODC2* transcript levels continued to decrease until 24 h after showing a slight increase at 1 h (Figure 5C). Expression of *SlAIH* gene generally remained unchanged up to 12 h of cold exposure with a measurable increase at 24 h (Figure 5D). Expression of *SlCPA* showed a dip at 6 h before increasing thereafter (Figure 5D). Transcript levels of *SlSAMDc1* increased significantly at 6 and 12 h of cold stress, reaching several-fold higher at 24 h of stress (Figure 5E). *SlSAMDc2* transcript levels increased several-fold by 6 h and continued to increase thereafter, reaching >150-fold at 24 h (Figure 5E). The expression of *SlSAMDc3* remained low until 12 h increasing significantly at 24 h (Figure 5E). The response of *SlSPDS1/2* genes to cold stress was seen only for *SlSPDS2* after 6 h with a notable increase at 24 h (Figure 5F). On the other hand, *SlSPMS* expression remained more or less similar for most time periods except for a slight increase at 12 h of cold stress (Figure 5G).

PA back conversion pathway gene transcripts of *SlPAO2* and *SlCuAO4-like* increased in response to cold treatment (Figure 5 H,I). Expression of *SlPAO4-like* gene also increased at 6, 12 and 24 h but the expression level was considerably lower than that observed for *SlPAO2* gene (Figure 5H). In comparison, the *SlCuAO4* gene expression remained unchanged during cold stress (Figure 5I). Significant differences for gene expression levels are listed in Appendix A.

The transcription factors, *C-repeat/DRE-Binding Factor* (*CBF*), are known to induce many cold stress-associated protein genes [20]. Therefore, transcript levels of three *CBF* gene family members (*CBF1, CBF2* and *CBF3*) as the cold stress marker genes were assessed. As was anticipated, all the three cold marker genes were strongly induced between 1 and 6 h of cold treatment, declining thereafter (Figure 5J–L).

#### 3.3.2. Different Forms of PAs Exhibit Strong Positive/Negative Pearson Correlations with One Another, with Transcript Levels of PA Metabolism, sHSPs and Other Genes during Cold Stress

Among the 72 total combinations for free, conjugated and bound PUT, SPD and SPM, 32 and 28 variables exhibited correlation coefficients of ≥ 0.8 and ≤ −0.8, respectively. PUT-F showed +ve *r* with PUT-B and SPD-B, and −ve *r* with SPD-F, SPM-F, SPD-C and SPM-B. SPD-F exhibited opposite relationship compared to PUT-F as it exhibited +ve *r* with SPM-F, SPD-C, SPM-C and SPM-B and −ve *r* with PUT-F, PUT-B and SPD-B. SPM-F showed +ve *r* for SPD-F, SPD-C and SPM-C and −ve *r* for PUT-F, PUT-B and SPD-B. Collectively, this analysis indicates the role of various PA forms in maintaining PA homeostasis during cold stress in tomato (Appendix A).

Under cold stress, PUT-F showed +ve *r* with ADC1, ODC1, and SAMDc2, while exhibiting −ve *r* with *ODC2*. Interestingly, *SPD-F* and SPM-F both exhibited identical *+ve*
*r* with *ODC2*, but −ve r with *ADC1*, *ODC1*, *SAMDC2* and *PAO2* in cold*-*stressed plants, suggesting coordinate regulation by free PAs (Appendix A).

To evaluate possible roles of different PA forms in regulating gene expression, we determined correlation coefficients for three marker CBFs (Appendix A). During cold stress, CBF2 and CBF3 showed +ve *r* and CBF1 −ve r with PUT-F, whereas CBF1 and CBF2 exhibited +ve *r* with SPM-B. All the three forms of SPD showed weak correlation to any of the CBFs. Among the other genes tested, the chloroplast located *psbA* showed *+ve*
*r* with Put-F and −ve *r* with SPD-F and SPM-F. *PEPC* expression also showed −ve *r* with *SPM-F* (Appendix A).

#### 3.3.3. Coordinated Regulation of PAs Biosynthesis/Catabolic and CBFs Protein Genes under Cold Stress

Several PA metabolism genes exhibited +ve r among themselves during the cold stress; these include *ADC1,2*; *ODC1; AHI1,2; AMDC1; SPDS; PAO2; PAO4-like* and *CUPAO4* (Appendix A). Thus, it would seem that both PA biosynthesis and catabolism are coordinately regulated under the cold stress. The three *CBF1-3* examined exhibited −ve *r* with *CPA,* suggesting inhibitory roles for these factors in PA metabolism (Appendix A). *ARG1* exhibited −ve *r* with all *Lhcb1-6,* whereas *ARG2* showed −ve *r* only with *Lhcb3-6*. Among the other genes that exhibited +ve *r* with PA metabolism genes included *OAT1, ICDH, Lhcb6, psbA* and *accD* (Appendix A). All *Lhcb1-6* showed coordinated regulation during cold stress. OAT1 and ICDH followed similar patterns showing +ve r with PEPC and ICDH but −ve r with Lhcb1-6 and CBF1, 2 (Appendix A).

### 3.4. Principal Component Analyses Highlights Differential Segregation of PA Metabolic, sHSPs and CBPs Genes for Heat versus Cold Stresses, Respectively

The multivariant analyses showed that heat and cold stresses exhibited opposite responses for many variables (Figure 6). The first two components, PC1 (48.16) and PC2 (36.20), accounted for over 84% variability (Figure 6). All heat-treated samples (variables) were present in +PC1 and +PC2 quadrants, whereas all the cold-treated ones were present in the opposite −PC1 +PC2 quadrants. None of the PAs metabolic genes nor free, bound or conjugated polyamines co-segregated with any of the heat-stressed samples suggesting limited or inhibitory roles of PAs in heat acclimation. In contrast and not surprisingly, the three small heat shock protein genes (*SlHSP17.6, SlHSP20.0,* and *SlHSP20.1*) co-segregated with the five ‘heat’ variables (increasing heat treatment) in +PC1 half, indicating their strong association with heat stress (Appendix A). For cold stress, ADC1, SAMDC2 and PAO2 along with CBF3 were present in the same quadrant (−PC1 +PC2) as the 12-C and 24-C cold stress samples, suggesting they are associated with cold stress (Appendix A). CBF1 and CBF2 co-segregated with early time (1-C and 6-C) cold stress samples in +PC1 −PC2 quadrant implicating their role in cold acclimation process.

Co-segregation of three cold-marker genes (*SlCBF1, SlCBF2,* and *SlCBF3*) with the four cold treatment samples confirms their function(s) during tomato response to cold stress. Interestingly, none of the three PA forms (free, bound or conjugated) co-segregated with cold stress samples in the same quadrant (−PC1, +PC2) indicating their limited direct role during the cold stress, but all of them segregated with the -PC1 half with the cold stress and not with the heat stress. Most of the PA metabolic (*ARG1, ARG2, ADC2, ODC1, ODC2, AIH1, CPA, SAMDC1, SAMDC3, SPD1, SPD2, SPMS, PAO4like, CuAO4, CuAO4like,* and *ICDH*) and other *Lhcb1-6, psbA* and *accD* genes evaluated were present in the −PC1 −PC2 quadrants indicating their limited role, if any, in these two stresses.

### 3.5. Cytoscape Analyses of Heat and Cold Stress Pooled Samples

To elucidate if any of the networks are shared by heat and cold stress, the Pearson correlation coefficient for the data pooled from heat and cold stresses were analyzed using Cytoscape program (Figure 7). This analysis resulted in two main and several small networks for the 41 genes and free, conjugated and bound forms of the three PAs. The largest network observed highlighted various forms of PAs, namely, PUT-F, PUT-C, SPD-F, SPD-C, SPD-B, SPM-F and SPM-B showing mostly strong −ve *r* or weak +ve *r* among themselves. Weak +ve *r* was observed for PUT-B with OAT1 and ARG1, and SPM-C with SPM-F. ARG1 exhibited strong −ve *r* with PUT-C, SPD-B and SPD-C and weak *r* with CPA whereas ODC1 exhibited strong −ve *r* with SPM-B. CPA, AHI1, CuAO4 and PAO4-like exhibited weak +ve *r* among themselves (Figure 7). The other strong network highlighted the PA metabolism genes *ADC1, ADC2, SAMDC1, SAMDC2* and *PAO2* showing strong +ve *r* among themselves. *SAMDC3* and *CuAO4-like* exhibited strong +ve *r* and *SPDS2* and *ODC1* exhibited weaker *r* with all other PA metabolic genes. *Lhcb1-6* except for *Lhcb2* resulted in their own network with relatively weak +ve *r* among each other. Three sHSPs and CBF1 exhibited their own networks, respectively, with weak +ve association with each other. We also analyzed Cytoscape networks separately for heat (Appendix A) and cold stresses (Appendix A). Heat stress exhibited a less elaborate and extensive network among various genes evaluated than the cold stress. Whereas many different forms of PAs exhibited strong −ve association with each other during both stresses, there were several noticeable differences among gene networks. Whereas PA metabolic genes generally exhibited weak +ve *r* during heat stress, strong +ve *r* was characteristic of cold stress (Appendix A). Taken together, these results provide evidence that PA gene expression differs remarkably during heat versus cold stress.

## 4. Discussion

Polyamines, as plant growth regulators, play important roles in the biology of plants and their longevity [28]. In recent years, research in determining the role(s) of different polyamines in plant responses to different stressors has intensified since environmental extremes affect the yield and quality of the produce. Here we have demonstrated the complexity of polyamine role(s) in relation to differential responses of tomato leaves when exposed to either heat or cold stress. Thus, differential expression of PA metabolism genes was apparent under heat versus cold stress with more impairment of PA metabolic genes taking place under heat stress. The data suggest a more pronounced role of PAs in cold stress, and yet an inability in coordinating their regulation under heat stress, which seems likely to be a reflection of their low plasticity to heat stress response.

Plants being sessile have developed different strategies to cope with the environmental stresses during their life cycle. Polyamines regulate a myriad of developmental and physiological processes including gene expression under both normal and stress conditions [28,69]. Metabolism of these biological amines has been implicated in developing tolerance or survival under harsh environmental conditions, such as heat, cold, drought and salinity stress [28,36,40,70]. Plant responses to temperature can be divided into four broad categories, including freezing (<−2 °C), chilly/low (−2 °C to 10 °C), optimum growth (15 °C to 25 °C) and high (>35 °C) temperatures. Except for the optimum temperature, the remaining temperatures are considered limiting for plant growth and crop yield, and thus stressful. Plants undergo myriad of morphological and physiological attributes so as to acclimate and survive under extreme temperatures [5]. In spite of a vast knowledge of plant ecosystems and plant survivability or not under harsh temperature conditions, the physiological and molecular mechanisms that could facilitate development, growth, reproductivity and yield of edible plants exposed to harsh environments have remained limited.

A multitude of studies have implicated PAs to play roles in plant response to temperature [37,71,72]. However, only in limited investigations has the response of isogenic plants at the same development stage to cold/heat stress and their biochemical responses been compared. Furthermore, most investigations are carried out at one time point or at best a few time points far apart from each other. Our studies presented here mark an effort to delineate differential responses of tomato leaves to two opposite stresses, heat versus cold, particularly to delineate their effects on biosynthesis and catabolism genes of PAs. We analyzed the kinetics of cold and heat stresses separately which yielded several novel results. Thus, it is shown herein that PA anabolic and catabolic genes in tomato leaves respond generally oppositely to heat versus cold stress (Figure 2, Figure 4 and Figure 6). For instance, free putrescene (PUT-F) levels decreased about 70% during the heat stress, while they continued to increase about 300% during the cold stress (Figure 2 and Figure 4). Over 30% decrease apparent in free spermidine (SPD-F) during heat stress was not found to be so under cold stress (Figure 2 and Figure 4), while free spermine (SPM-F) increased during heat stress but exhibited little change during cold stress (Figure 2 and Figure 4). There was also similarity in the patterns of free PUT, SPD and SPM steady-state levels and conjugated forms of PUT, SPD and SPM under heat versus cold stress. Moreover, increase in conjugated PUT (PUT-C) during cold stress was opposite to their decrease during heat stress (Figure 2 and Figure 4). The levels of bound SPD (SPD-B) remained high throughout the cold stress but declined precipitously during heat stress (Figure 2 and Figure 4). Several reports have indicated that cold temperatures induce different levels of PA accumulation. For example, PUT content increased in tomato seedlings under low-temperature conditions [73]. Thus, our findings suggest that pools of free and conjugated PAs are responsible for maintaining the PAs homeostasis under the two stresses investigated. This interpretation has significant bearing on the recovery of crops under heat or cold stress. It is likely that plants can recover from cold stress to a greater extent than the heat stress.

In regard to PA synthesis genes, two *ADCs* (*SlADC1* and *2*), one *ODC* (*SlODC2*), and two *SAMDc* (*SAMDc1* and *SAMDc3*) genes were upregulated during heat stress. The *SlSPDS2* gene expression was induced under cold stress which is consistent with the finding of an earlier study [73]. Furthermore, cold-responsive elements in the promoters of *SlADC1* and *SlODC1, 2* were previously reported [57]. Interestingly, higher expression of *SlPAO2* and *SlCuAO4-like* genes was noticeable. The levels of different polyamines were found in sync with transcript data, for example, a clear increase in the PUT levels was apparent as the cold stress period increased. To our knowledge, this is the first report showing that extreme temperature regimes (heat versus cold) exhibit opposite response to PUT accumulation in tomato leaf tissues.

Other findings of the data presented here are related to a differential response of PA biosynthesis genes to heat versus cold stress. Namely, transcript levels of *ARG1; AHI1,2; SAMDC1; SAMDC2; SPDS1;* and *SPDS2* increased in response to heat stress but their transcript levels were down regulated during cold stress. Oppositely, transcript levels of *ADC1, ADC2, ODC11, ODC2,* and *SPMS* declined during heat stress but increased during cold stress. Previously, an increased SPD content together with an increase in ADC enzyme activity was observed in cucumber plants under cold treatment [74]. Cold treatment was found to alter ODC enzyme activity and ODC expression [73]. Most of the earlier studies have focused primarily on PA biosynthesis while PAs catabolism/back conversion is less studied in tomato fruit ripening [48] and grape ripening [75]. Recently, PA biosynthesis [56] and PAO mediated catabolism [57] under stress has been reported but changes in PAs levels were not determined in these investigations. In conclusion, whereas *PAO4-like* and *CuAO4* were up regulated in heat stress, *PAO2* and *CuAO4-like* were upregulated under cold stress, contrasting their roles in the two stresses studied here. The multivariant analyses (Figure 6) gives further support to our conclusion that polyamine gene expression and metabolism under heat versus cold stress exhibit unique and opposite responses.

## Figures and Tables

**Figure 1 cells-09-01749-f001:**
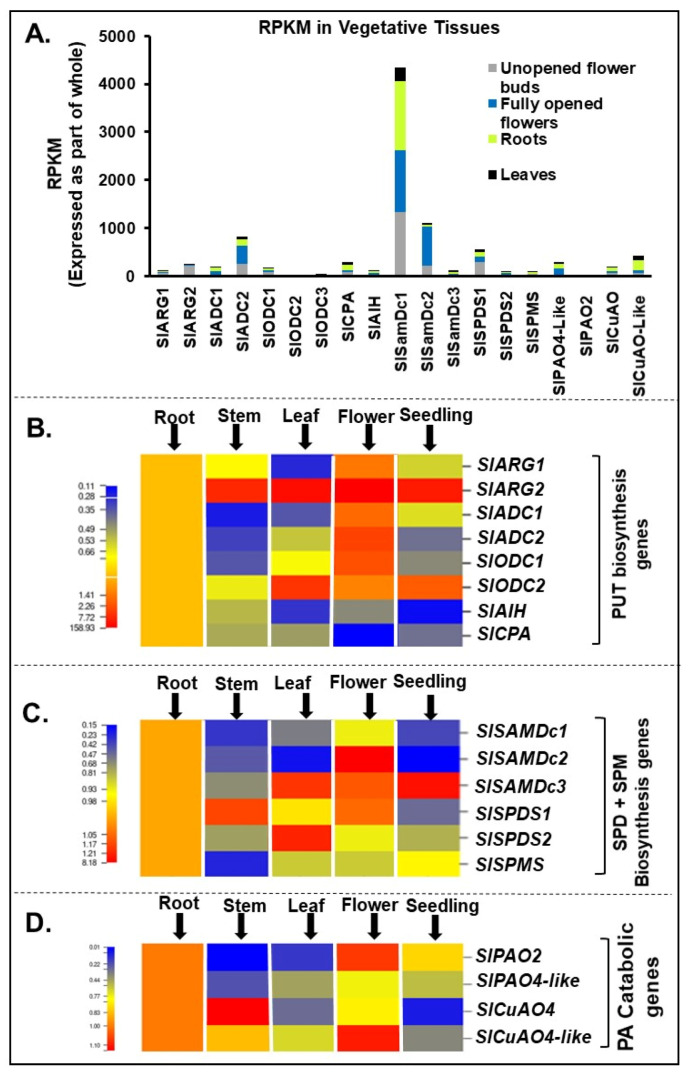
Spatiotemporal gene expression of tomato (*Solanum lycopersicum* L.) polyamines (PA) metabolic pathway genes. (**A**) Global transcriptome analysis of tomato vegetative tissues (flower-buds, open flower, leaf and roots) reveals differential abundance of polyamine biosynthesis genes *SlARG1, 2*, *SlADC1, 2 SlODC1, 2, SlAIH* and *SlCPA* and catabolic enzymes encoding genes *SlPAO2 and SlPAO4-Like; SlCuAO and SlCuAO4-like*. (**B**) qRT-PCR based gene expression analysis of tomato vegetative tissues (root, stem, leaf, open flower and seedling tissues) reveals differential abundance of putrescine (PUT) biosynthesis genes *SlARG1, 2*, *SlADC1, 2 SlODC1, 2, SlAIH* and *SlCPA*. (**C**) Spermidine (SPD) and spermine (SPM) biosynthesis genes *SlSAMDc1, 2, 3, SlSPDS1, 2* and *SlSPMS*. (**D**) PA catabolic pathway genes *SlPAO2 and SlPAO4-Like; SlCuAO and SlCuAO4-like*. A minimum of three plants were taken for collection of vegetative tissues for each biological replicate and mean data points and standard error were derived from three biological replicates. Statistical significance (*p*-value) was derived using Graph prism pad program via Tukey test/t-test. *SlUBI3* and *SlTIP41* were used as reference genes.

**Figure 2 cells-09-01749-f002:**
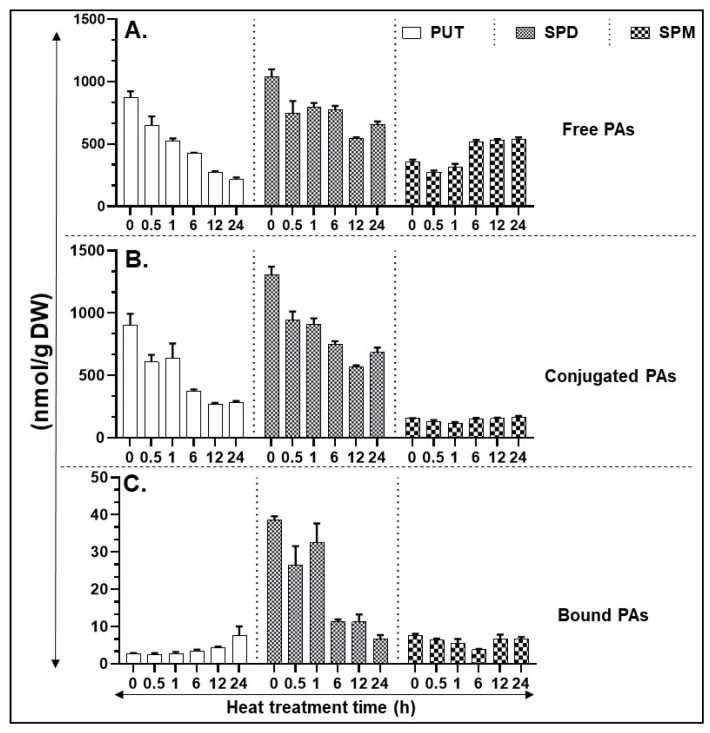
HPLC (high pressure liquid chromatography) based quantification of polyamines (PA) in heat-stressed tissues. (**A**) Free PAs—PUT, SPD, and SPM. (**B**) Conjugated PAs—PUT, SPD, and SPM. (**C**) Bound PAs—PUT, SPD, and SPM. A minimum of three leaves were excised from three plants for each biological replicate. Mean data points and standard error of a minimum of three biological replicates are shown. Statistical significance (*p*-value) was derived using Graph prism pad program via Tukey test/t-test.

**Figure 3 cells-09-01749-f003:**
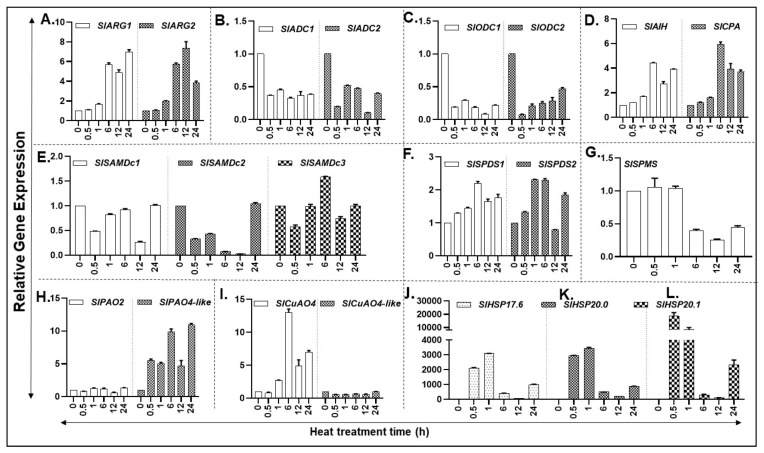
Heat-induced expression of PA metabolic pathway encoding genes in tomato plants exposed to 42 °C temperature. (**A**–**D**) qRT-PCR based relative gene expression analysis of tomato leaf tissue reveals differential abundance of putrescine (PUT) biosynthesis genes (*SlARG1, 2*, *SlADC1, 2 SlODC1, 2, SlAIH* and *SlCPA*); (**E**–**G**) Spermidine (SPD) and Spermine (SPM) biosynthesis genes (*SlSAMDc1, 2, 3, SlSPDS1, 2* and *SlSPMS*); (**H**,**I**) PA catabolic pathway genes (*SlPAO2* and *SlPAO4-Like; SlCuAO and SlCuAO4-like*); (**J**–**L**) Heat shock protein genes *HSP17.6, 20.0* and *20.1*. A minimum of three leaves were excised from three plants for each biological replicate. Mean data points and standard error of a minimum of three biological replicates are shown. Statistical significance (*p*-value) was derived using Graph prism pad program via Tukey test/t-test. *SlUBI3* and *SlTIP41* were used as reference genes.

**Figure 4 cells-09-01749-f004:**
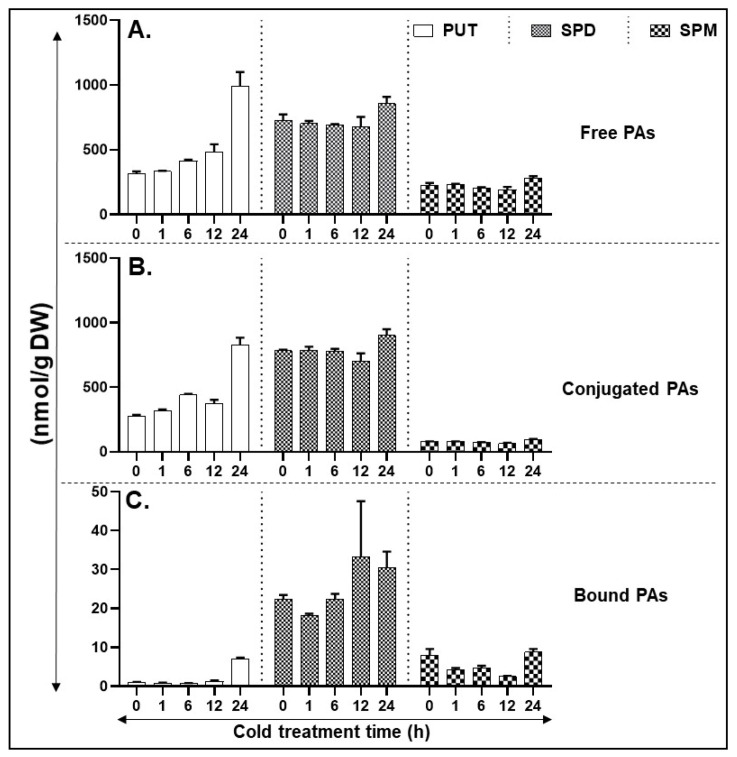
HPLC (High pressure liquid chromatography) based quantification of polyamines (PA) levels in cold-stressed tissues. (**A**) Free PAs—PUT, SPD, and SPM. (**B**) Conjugated PAs—PUT, SPD, and SPM. (**C**) Bound PAs—PUT, SPD, and SPM. A minimum of three leaves were excised from three plants for each biological replicate. Mean data points and standard error of a minimum of three biological replicates are shown. Statistical significance (*p*-value) was derived using Graph prism pad program via Tukey test/t-test.

**Figure 5 cells-09-01749-f005:**
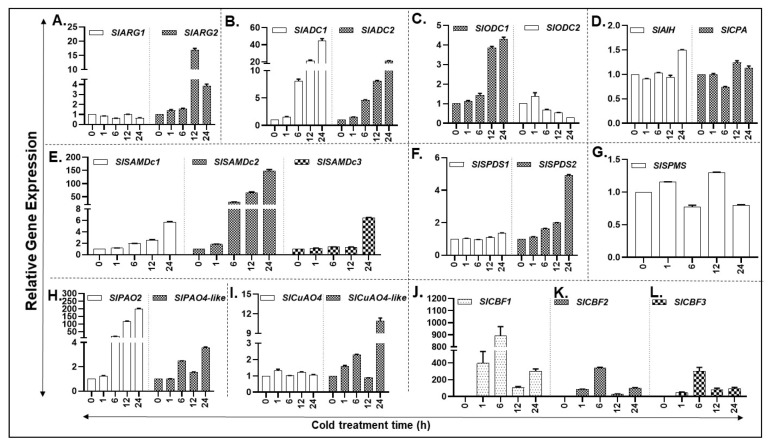
Cold-induced expression of PA metabolic pathway genes in tomato plants exposed to 4 °C temperature. (**A**–**D**) qRT-PCR led relative gene expression analysis of tomato leaf tissue reveals differential abundance of putrescine (PUT) biosynthesis genes (*SlARG1, 2*, *SlADC1, 2 SlODC1, 2, SlAIH* and *SlCPA*); (**E**–**G**) Spermidine (SPD) and Spermine (SPM) biosynthesis genes (*SlSAMDc1, 2, 3, SlSPDS1, 2* and *SlSPMS*); (**H**,**I**) PA catabolic pathway genes (*SlPAO2 and SlPAO4-Like; SlCuAO and SlCuAO4-like*); (**J**–**L**) Cold-responsive genes *CBF1, 2* and *3*. A minimum of three leaves were excised from three plants for each biological replicate. Mean data points and standard error of a minimum of three biological replicates are shown. Statistical significance (*p*-value) was derived using Graph prism pad program via Tukey test/t-test. *SlUBI3* and *SlTIP41* were used as reference genes.

**Figure 6 cells-09-01749-f006:**
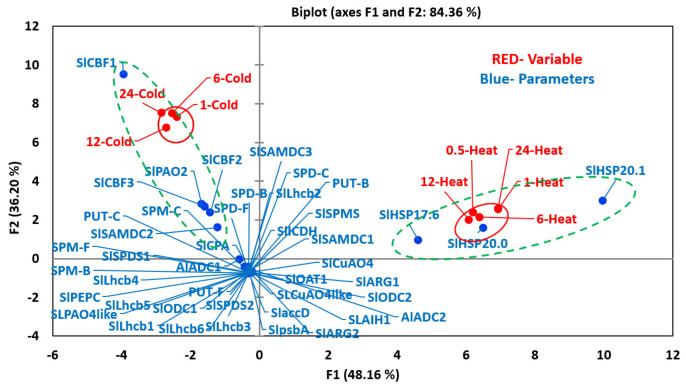
PCA analyses of variables and parameters during heat and cold stresses. The variables included seedlings treated to increasing time period after applying heat (0.5, 1, 6, 12 and 24 h) and cold (1, 6, 12 and 24 h) stresses. The parameters included levels of free (F), conjugated (C) and bound (B) PUT, SPD and SPM (PUT-F, SPD-F, SPM-F, PUT-C, SPD-C, SPM-C, PUT-B, SPD-B and SPM-B), transcript levels of PAs biosynthesis and catabolizing enzymes genes, heat and cold stress maker genes and photosynthetic-machinery-related genes. For PCA analyses, all values for active and variable parameters were normalized as per cent of that at zero time (before the initiation of respective stresses) values. Other details are the same as stated in the Materials and Methods section. Genes’ abbreviations are given in the Appendix A. PCA analyses was performed using XLSTAT (https://www.xlstat.com).

**Figure 7 cells-09-01749-f007:**
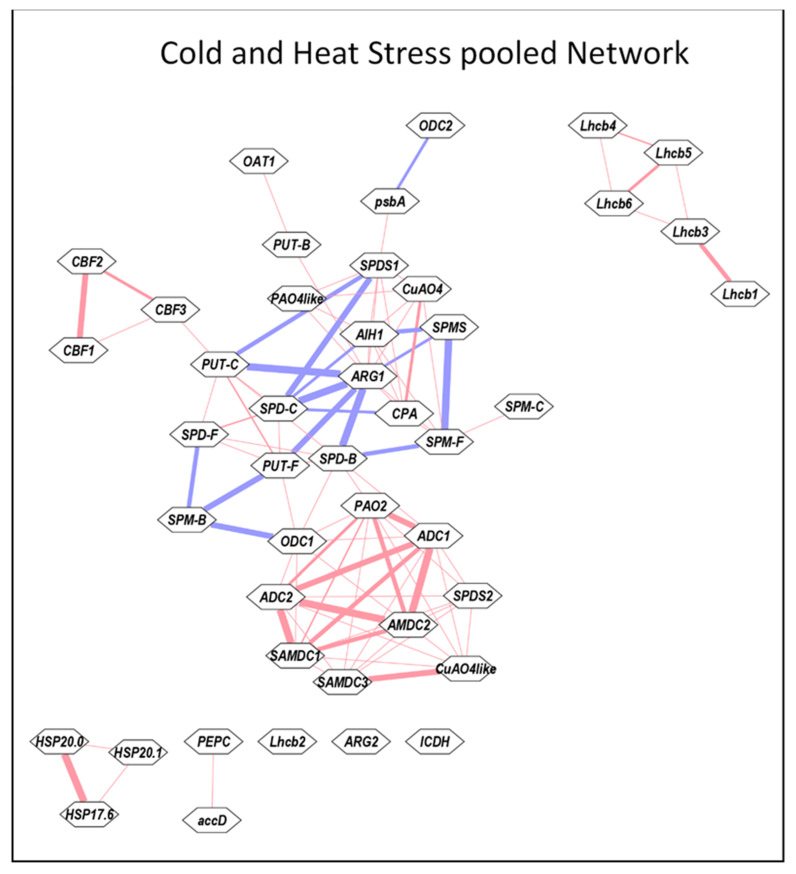
Cytoscape analyses of Pooled data from heat and cold stresses. All values for the active and the variable parameters were normalized as % initial 0 time point. Before determining correlation coefficient *r*, data for each parameter was pooled individually from heat and cold stresses. The correlation coefficients were determined using Microsoft EXCEL. These data were analyzed using the Expression Correlation app of the Cytoscape program [65]. Gene abbreviations are the same as in the Appendix A.

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
