# Peer review of "Polyamines and Their Biosynthesis/Catabolism Genes Are Differentially Modulated in Response to Heat Versus Cold Stress in Tomato Leaves (Solanum lycopersicum L.)"

_cells, 2020, doi:10.3390/cells9081749_

Round 1

Reviewer 1 Report

The manuscript "Polyamines and their biosynthesis/catabolism genes are  differentially modulated in response to heat versus cold stress in tomato leaves (Solanum lycopersicum L.)" is a very interesting work and the authors present a vast number of data that is helpful to go deeper inside into the role of PA in plants subjected to different stressful conditions. 

I think the paper is well written and I have only a few notes:

  • line 19: "thwo";
  • line 61: please write "programmed cell deadh (PCD)";
  • lines 69, 78: please write the extended name the first time you use it in the text and then use the abbreviation;
  • line 85: please check the references after [56];
  • line 126: there is a "[" before "international tomato..." but I couldn't find any "]";
  • line 258: please delete one "dot" at the end of the sentence;
  • line 368: figure 5 caption. only in this caption you use the bold style for letters;
  • lines 382-386: please check the style;
  • lines 510, 511: the reference and table T1 are in bold type, please correct.

Author Response

The manuscript "Polyamines and their biosynthesis/catabolism genes are  differentially modulated in response to heat versus cold stress in tomato leaves (Solanum lycopersicum L.)" is a very interesting work and the authors present a vast number of data that is helpful to go deeper inside into the role of PA in plants subjected to different stressful conditions. 

I think the paper is well written and I have only a few notes:

  • line 19: "thwo";
  • line 61: please write "programmed cell deadh (PCD)";
  • lines 69, 78: please write the extended name the first time you use it in the text and then use the abbreviation;
  • line 85: please check the references after [56];
  • line 126: there is a "[" before "international tomato..." but I couldn't find any "]";
  • line 258: please delete one "dot" at the end of the sentence;
  • line 368: figure 5 caption. only in this caption you use the bold style for letters;
  • lines 382-386: please check the style;
  • lines 510, 511: the reference and table T1 are in bold type, please correct.

Answer: Thanks for your constructive comments. We have taken care of all of your suggestions and inputs. Changes were made using track changes at the respective places.

Reviewer 2 Report

The authors study quantified polyamines and the expression of genes involved in polyamine metabolism in tomato suggested to cold or heat.

They have nice data, that they use for Pearson correlation and PCA analysis.

The take home message is that polyamine metabolism is likely to be more relevant for cold stress than for heat stress.

While it is a nice study, it is mainly descriptive and I do not know if it is important enough to be published in a journal of 5.656 impact factor. I leave the editor to decide.

Please find under my comments.

Lines 15/16 : delete sentence starting with « Here, we …”

Line 32: what do you mean by “more impairment of pA metabolism”. How do you quantify metabolism?

Line 33. How can you speak of their inability to coordinate regulation. You have no data for that.

Lines 44/46: repetitive

Line 61, full name for PCD

Line 72. Rephrase; delete ODC and ADC. Speak of the Arabidopsis exception only after, once you have presented the pathway.

Could there be a figure to present the pathway?

Line 75, what does arginase catalyze?

Line 89; rephrase “metabolic pathway gene expression”

Line 99 and after; what is light intensity, light regime and temperature for normal condition

Line 101: explain; you transplant what? From where to where?

Line 107: what is DAT

Line 112 and after: what age for cold treatment? 28 DAT also? Indicate.

Line 123: “PA gene”

Line 125: italics for plant species

Line 123 and after: is it two approaches or one approach with two consecutive steps?

Line 199: what is the interest of panel A since then you have your own qPCR data. I would delete.

Line 190: what is the interest and relevance to compare all genes together? Only makes sense to compare genes within one family.

Line 252, figure 2. Why no asterisks or letters for stat. significance? Why only in supplementary?    Idem figures 3, 4 and 5

Line 296; what is “+ ve r”? you define r the sentence before but no  “+ ve r”?

Line 196 : define B, C and F. You have not used before I think.

Line 419. I do not understand your PCA figures. Usually you have parameters and variables. You have blue and red, points and lines. You write “active variables” and “active observations” but I do not get what you mean by that.  

You write in caption; “The variable parameters included levels of free (F), 437 conjugated (C) and bound (B) PUT, SPD and SPM (PUT-F, SPD-F, SPM-F, PUT-C, SPD-C, SPM-C, 438 PUT-B, SPD-B and SPM-B).” but I understand the expression level of stress markers genes were also used as parameters;

So clearly define in the text what is used as variable and what is used as parameters.

You shold write cold and not C and heat and not H because if not it is confusing.

Author Response

Reviewer 2

The authors study quantified polyamines and the expression of genes involved in polyamine metabolism in tomato suggested to cold or heat. They have nice data, that they use for Pearson correlation and PCA analysis. The take home message is that polyamine metabolism is likely to be more relevant for cold stress than for heat stress. While it is a nice study, it is mainly descriptive and I do not know if it is important enough to be published in a journal of 5.656 impact factor. I leave the editor to decide.

Please find under my comments.

Lines 15/16 : delete sentence starting with « Here, we …”

Answer: Done.

Line 32: what do you mean by “more impairment of PA metabolism”. How do you quantify metabolism?

Answer: The sentence was modified and re-written

Line 33. How can you speak of their inability to coordinate regulation. You have no data for that.

Answer: The sentence was restructured. 

Lines 44/46: repetitive

Answer:. We would like to keep this sentence as is.

Line 61, full name for PCD.

Answer: Thanks for bringing this to our attention. Done.

Line 72. Rephrase; delete ODC and ADC. Speak of the Arabidopsis exception only after, once you have presented the pathway.

Could there be a figure to present the pathway?

Answer: We have made some changes. The PA pathway and exception with Arabidopsis not having ODC gene is well known in the literature.  Therefore, we believe showing a pathway is not a great idea.

Line 75, what does arginase catalyze?

Answer: Arginase catalyzes arginine to ornithine and urea (Shi and Chan, 2013).

Line 89; rephrase “metabolic pathway gene expression”

Answer: Sentence rephrased.

Question: Line 99 and after; what is light intensity, light regime and temperature for normal condition

Answer: We have mentioned this in methods related to specific treatments. Please see section 2.2 from line 107-116.

Question: Line 101: explain; you transplant what? From where to where?

Answer: As is known, there are less chances of achieving 100% seed germination and not all seeds germinate at the same time point. To circumvent this, we sow more seed than the number needed. After 8-10 days, two cotyledonary leaves bearing seedlings, are transplanted to bigger pots to set up the experiment. Thereafter, the seedlings are grown for 4 weeks [28DAT (days after transplant)] until specific stress was imposed.

Question: Line 107: what is DAT

Answer: DAT=days after transplant

Question: Line 112 and after: what age for cold treatment? 28 DAT also? Indicate.

Answer: 28 DAT old plants were subjected to cold treatment. We added this in method section under section 2.2.

Question: Line 123: “PA gene”

Answer: We wrote the full name.

Question: Line 125: italics for plant species

Answer: We italicized plant species names.

Question: Line 123 and after: is it two approaches or one approach with two consecutive steps?

Answer: BLASTP search with known sequences is different than Hidden Markov Model (HMM) search used through HMMER. HMM search exclusively uses conserved known gene family protein domains with definite annotation. So, it is not two consecutive steps. These are two different approaches to find new homologues in newer species where not known previously. 

Question: Line 199: what is the interest of panel A since then you have your own qPCR data. I would delete.

Answer: Panel A in Figure 1 is important to show and match our data with publicly available transcriptome data. It reveals data accuracy and also helps readers.

Question: Line 190: what is the interest and relevance to compare all genes together? Only makes sense to compare genes within one family.

Answer: This is a very important question and we think that publishing data in bits and pieces is not a good science. Some researchers like to publish polyamine biosynthesis and catabolism separately. We found it more relevant to investigate together genes encoding for the PA pathway as much as possible.

Question: Line 252, figure 2. Why no asterisks or letters for stat. significance? Why only in supplementary?    Idem figures 3, 4 and 5

Answer: The idea to provide significant differences separately as a supplementary table is our idea to avoid figures being messy and complicated. Our supplementary file contains significance values for each data point which allows readers to look at all the data set differential changes at one place.

Question: Line 296; what is “+ ve r”? you define r the sentence before but no  “+ ve r”?

Answer: + ve stands for ‘positive’ – done (Defined in the revised version).

Question: Line 196: define B, C and F. You have not used before I think.

Answer: Done.

Question: Line 419. I do not understand your PCA figures. Usually you have parameters and variables. You have blue and red, points and lines. You write “active variables” and “active observations” but I do not get what you mean by that.  

You write in caption; “The variable parameters included levels of free (F), 437 conjugated (C) and bound (B) PUT, SPD and SPM (PUT-F, SPD-F, SPM-F, PUT-C, SPD-C, SPM-C, PUT-B, SPD-B and SPM-B).” but I understand the expression level of stress markers genes were also used as parameters;

So clearly define in the text what is used as variable and what is used as parameters.

You should write cold and not C and heat and not H because if not it is confusing.

Answer: Done. A new figure was made instead.

Thank you for your incisive review. We appreciate your input.

Reviewer 3 Report

This manuscript is well written and contributes significant information to scientific literature on the role of PAs in stress responses of tomato.

Only minor changes are suggested for this manuscript:

  1. Line 61: Spell out the full term for PCD before giving the abbreviation in parentheses.
  2. Line 64: Provide the abbreviated term for all PAs.

Since genes for the PA pathways are known and listed, their sequences should be easily found in tomato whole-genome sequencing database. The authors could perform a bioinformatic study to map the physical locations of all PA pathway genes. The manuscript of such a study would be suitable for the special issue "Mapping Abiotic Stress-Tolerance Genes in Plants 2020" of IJMS, which is being set up with a December 31, 2020 deadline for submission. 

Author Response

Reviewer 3

This manuscript is well written and contributes significant information to scientific literature on the role of PAs in stress responses of tomato.

Only minor changes are suggested for this manuscript:

  1. Line 61: Spell out the full term for PCD before giving the abbreviation in parentheses.
  2. Line 64: Provide the abbreviated term for all PAs.

Since genes for the PA pathways are known and listed, their sequences should be easily found in tomato whole-genome sequencing database. The authors could perform a bioinformatic study to map the physical locations of all PA pathway genes. The manuscript of such a study would be suitable for the special issue "Mapping Abiotic Stress-Tolerance Genes in Plants 2020" of IJMS, which is being set up with a December 31, 2020 deadline for submission. 

Answer: Thanks for your constructive comments. PA pathway gene information and their locus identities are given in Supplementary Table 1. We have taken care of your suggestions and inputs. We have submitted this manuscript to ‘Cells’ journal based on an invitation to Dr. Autar Mattoo and Dr. Avtar Handa from guest editors of this special section “Plant polyamines in plant stress tolerance”. We will submit our next manuscript to your suggested IJMS journal issue